# Disease risk analysis in sea turtles: A baseline study to inform conservation efforts

Narges Mashkour[1]*, Karina Jones[1,2]◉, Sara Kophamel[1]◉, Teresa Hipolito[1], Shamim Ahasan[1,3], Grant Walker[4,5], Richard Jakob-Hoff[6,7], Maxine Whittaker[1], Mark Hamann[8], Ian Bell[9], Jennifer Elliman[1], Leigh Owens[1], Claire Saladin[10,11], Jose Luis Crespo-Picazo[12], Brett Gardner[13,14], Aswini Leela Loganathan[15], Rachel Bowater[1], Erina Young[16], David Robinson[17], Warren Baverstock[17], David Blyde[18], Duan March[19,20], Maryam Eghbali[21], Maryam Mohammadi[22], Daniela Freggi[23], Jane Giliam[24], Mike Hale[25], Nicholas Nicolle[14], Kevin Spiby[14], Daphne Wrobel[26], Mariluz Parga[27], Asghar Mobaraki[28], Rupika Rajakaruna[29], Kevin P. Hyland[30], Mark Read[31], Ellen Ariel[1]

1 College of Public Health, Medical and Veterinary Sciences James Cook University, Townsville, Australia, 2 College of Medicine and Dentistry, James Cook University, Townsville, Australia, 3 Faculty of Veterinary and Animal Science, Hajee Mohammad Danesh Science & Technology University, Dinjapur, Rangpur, Bangladesh, 4 North East Sea Turtles, Charlotteville, Tobago, Trinidad and Tobago, 5 Institute of Biodiversity Animal Health and Comparative Medicine, University of Glasgow, Glasgow, Scotland, 6 New Zealand Centre for Conservation Medicine, Auckland Zoo, Auckland, New Zealand, 7 School of Veterinary and Biomedical Sciences, Murdoch University, Murdoch, Australia, 8 College of Science and Engineering, James Cook University, Townsville, Australia, 9 Faculty of Health and Behavioural Sciences, University of Queensland, Brisbane, Australia, 10 Reserve Naturelle de Saint Barthelemy, Gustavia, Saint Barthelemy, 11 FWI/Reserve Naturelle de Saint Martin, Saint Martin, French West Indies, 12 Veterinary Services & Research Department, Fundación Oceanogràfic, Avanqua Oceanogràfic-Ágora, Valencia, Spain, 13 Australia Zoo Wildlife Hospital, Beerwah, Queensland, Australia, 14 Two Oceans Aquarium, Cape Town, South Africa, 15 Biotechnology Research Institute, Universiti Malaysia, Sabah, Malaysia, 16 Conservation Medicine Program School of Veterinary and Life Sciences, College of Veterinary Medicine Murdoch University, Perth, Western Australia, 17 The Aquarium & Dubai Turtle Rehabilitation Project, Burj Al Arab, Dubai, United Arab Emirates, 18 Sea World, Gold Coast, Queensland, Australia, 19 National Marine Science Centre & Centre for Coastal Biogeochemistry Research, School of Environment, Science and Engineering, Southern Cross University, Coffs Harbour, NSW, Australia, 20 Dolphin Marine Rescue Animal Rehab Trust, Coffs Harbour, New South Wales, Australia, 21 Ideh no doostdar_E- Hormozgan Ecotourism and NGO group, Iran, 22 Department of Environment, Kish, Iran, 23 Sea Turtle Rescue Centre, Lampedusa, Italy, 24 The Ark Animal Hospital, Northern Territory, Australia, 25 Yuku Baja Muliku Ranger Base, Archer Point, Australia, 26 Fundação Pró-TAMAR, Rua Professor Ademir Francisco s/n–Barra da Lagoa, Florianópolis–SC, Brazil, 27 SUBMON—Marine Environmental Services, Barcelona, Spain, 28 Department of the Environment, Wildlife and Aquatic Affairs Bureau, Tehran, Iran, 29 University of Peradeniya, Peradeniya, Sri Lanka, 30 Wildlife Protection Office, Dubai, United Arab Emirates, 31 Field Management Unit, Great Barrier Reef Marine Park Authority, Queensland, Australia

◉ These authors contributed equally to this work.
* narges.mashkour@my.jcu.edu.au

**Data Availability Statement:** All relevant data are within the manuscript and its Supporting Information files.

**Funding:** The international workshops and the publication fee is supported by James Cook University higher degree by research enhancement

## Abstract

The impact of a range of different threats has resulted in the listing of six out of seven sea turtle species on the IUCN Red List of endangered species. Disease risk analysis (DRA) tools are designed to provide objective, repeatable and documented assessment of the disease risks for a population and measures to reduce these risks through management options. To the best of our knowledge, DRAs have not previously been published for sea turtles, although disease is reported to contribute to sea turtle population decline. Here, a comprehensive list of health hazards is provided for all seven species of sea turtles. The possible risk these hazards pose to the health of sea turtles were assessed and "One

scheme. The first author has applied for this grant and was awarded the grant to carry out the research.

**Competing interests:** NO authors have competing interests.

Health" aspects of interacting with sea turtles were also investigated. The risk assessment was undertaken in collaboration with more than 30 experts in the field including veterinarians, microbiologists, social scientists, epidemiologists and stakeholders, in the form of two international workshops and one local workshop. The general finding of the DRA was the distinct lack of knowledge regarding a link between the presence of pathogens and diseases manifestation in sea turtles. A higher rate of disease in immunocompromised individuals was repeatedly reported and a possible link between immunosuppression and environmental contaminants as a result of anthropogenic influences was suggested. Society based conservation initiatives and as a result the cultural and social aspect of interacting with sea turtles appeared to need more attention and research. A risk management workshop was carried out to acquire the insights of local policy makers about management options for the risks relevant to Queensland and the options were evaluated considering their feasibility and effectiveness. The sea turtle DRA presented here, is a structured guide for future risk assessments to be used in specific scenarios such as translocation and head-starting programs.

## 1. Introduction

The International Union for Conservation of Nature (IUCN) has listed six of the seven sea turtle species on the IUCN Red List of endangered species while the seventh species, the flatback turtle (*Natator depressus*), is reported as "Data Deficient" [1]. Over the past 100 years, the world population of sea turtles has declined due to direct and indirect human interventions [2]. Disease is likely a contributing or primary factor in sea turtle deaths and poses challenges to conservation programs [3], but due to a number of factors, including the challenges of sampling wild marine animals in remote areas, incidences are generally underreported [4].

It is particularly difficult to capture a sea turtle with clinical signs in the wild as sea turtles are often hard to locate and difficult to access in remote areas [5]. Postmortem examination provides the most robust opportunity to identify diseases and their aetiology. Unfortunately, the difficulty of retrieving carcasses in the wild, as well as postmortem changes, can complicate the process of making a reliable diagnosis [6]. In addition, the results of such studies would not aid in determining the rate of morbidity versus mortality. An alternative way to investigate wildlife disease is to conduct controlled experimental studies, but due to their endangered status, such studies are difficult to justify for sea turtles [7].

Advanced biodiversity monitoring techniques for sea turtles conservation is gaining popularity including alternative and efficient techniques such as environmental DNA [8, 9]. Screening eDNA is considered to be a non-invasive way to assess the population dynamics of sea turtles [10], along with investigating the causation and progression of diseases such as fibropapillomatosis [11]. Satellite telemetry is another advanced tool successfully used in sea turtle population assessment and marine parks management [12].

However such technologies are still evolving and face challenges that make them not globally accessible; eDNA for instance, requires finances, advanced laboratory equipment and skilled operators to analyse and interpret the results, otherwise it may not provide accurate information [13]. The satellite deployment sites are biased towards for example North America rather than Africa due to the cost of developing, maintaining and utilising these techniques [14].

Scientists have validated the methods used for health assessment of other animals in sea turtles [3] and applied these procedures for sea turtle health and rehabilitation [15]. Despite this, it is still challenging in some instances to diagnose the cause of disease or death in sea turtles [16] and prevention and control measures are therefore not fully achievable [17].

The limitations and uncertainties of wildlife disease assessment call for structured, evidence-based approaches to inform management and reduce the risk of diseases, where disease drivers and their contribution to other threats can be defined. Wildlife Disease Risk Analysis (DRA) is most effective when taking a multidisciplinary approach involving scientists, clinicians and relevant decision makers to develop rational, effective and unbiased conclusions for wildlife health surveillance in support of conservation strategies.

The latest DRA manual was published by the World Organisation for Animal Health (OIE) and IUCN Species Survival Commission in 2014. The manual addresses different scenarios for endangered species and translocating them for conservation purposes and enables the pros and cons of these actions to be thoroughly investigated [18]. In order to accommodate the unique biology of sea turtles, the DRA process as described in this manual requires certain modifications to realistically articulate with situations such as translocating animals or investigating the risks of disease for a population in its normal habitat. A 2015 study describes a systematic approach to investigate disease-related population decline without confining the assessment to a particular scenario or location [6]. This method is a modified version of a DRA based on epidemiological principles [6] for any declining wildlife population. A successful DRA considers the study population in the context of the environment.

In the 1960's, Calvin Schwabe coined the term "*One Medicine*" which then extended to *"One Health"* that takes into account the inter-dependent health of humans, livestock and wildlife [19, 20]. One Health is an all-inclusive collaboration between public health, animal health and environmental specialists as well as communities and social scientists, through a transdisciplinary approach, to sustain the world's health [21]. The founding belief behind promoting One Health is the interconnected health of humans, animals and the environment. Approximately 75% of human infectious diseases are zoonotic, or in other words, are caused by multi-host pathogens carried by animals [19]. Unsustainable degradation of the environment by humans, toxins and chemical contaminants are also known to enhance the rate of emerging diseases in people, wildlife and livestock [22, 23]. Humans are also contributing to pressure on wildlife by the increasing demands for meat protein and subsequent habitat degradation [19].

Disease affects not only a population, but also the habitat, the other animals and humans that share it and *vice versa*. In the context of One Health, green turtles (*Chelonia mydas*) are particularly important due to their longevity and fidelity to a near-shore foraging site [24, 25]. Their continuous and long-term residency in a given location makes them good sentinels for local environmental health [26] and thereby function as marine 'ambassadors' for One Health.

There are currently no published reports on DRA for sea turtles and this gap compromises strategies presently implemented to address sea turtle conservation action such as disease control, clutch translocations and hatchery establishment. In this study, both DRA models described by Jakob-Hoff *et al.* [18] and Pacioni *et al.* [6] were integrated to highlight how these guidelines can be used to develop a DRA for sea turtles. The purpose of this study is to provide a baseline DRA which should serve as an example of this process for future, case-specific studies aiming to inform management decisions. The interrelated health of sea turtles, marine and terrestrial animals, humans and the environment were also addressed to define One Health factors.

## 2. Methods

The process of a DRA is outlined in Fig 1. Briefly, DRA organisers define a specific scenario for a wildlife population, for example translocating a clutch of sea turtle eggs from A to B (Step 1. Problem description). Then, published literature and unpublished reports about the hazards are collected and a group of experts are invited to review the information. This collection of comprehensive knowledge enables identification of hazards to the population under consideration (Step 2. Hazard identification). Assessing the knowledge of likelihood and consequences for each hazard, ideally conducted as a workshop with invited experts, will help to prioritise the need for research or surveillance strategies (Step 3. Risk assessment). Following a structured risk assessment, the prioritised health hazards or risks will be presented to a group of stakeholders who will review management options and the use of these options based on an assessment of their feasibility and effectiveness (Step 4. Risk management). The final step (Step 5. Implementation and review) is focused on finding the possible errors in executing the solutions suggested in the process [18].

### 2.1. Problem description

The larger the spatial scale of the area of interest, the harder it is to describe the risks and apply management. For this reason, "Management units" need to be defined alongside the problem description. The DRA must focus on localised scenarios such as translocating a clutch of eggs from A to B, establishing a turtle hatchery in location X or the outbreak of a bacterial or parasitic infectious disease in a rookery. However, as this DRA is a guideline for future researchers and managers to facilitate realistic risk management, defining a problem description for the

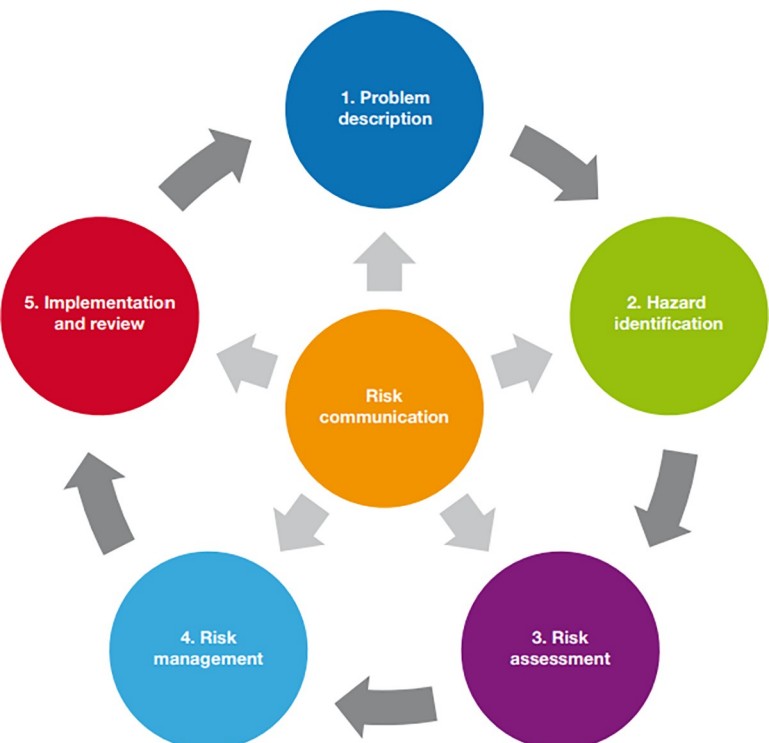

**Fig 1. Steps in the disease risk analysis process, reproduced from the DRA manual published by OIE and IUCN (2014).**

present study involving the global population of sea turtles would be erroneous. Although, the population decline of sea turtles and the difficulties in disease diagnosis suggest that the problem can be described as: "Certain infectious or non-infectious diseases are likely to contribute to sea turtle population decline" this is not specific enough to make the risk management achievable. To capture all of the expertise in this field, we have refrained from defining a problem description. However, in the interest of this guide, one example with specific problem description is given in S1 Appendix in S1 File.

## 2.2. Hazard Identification

A "hazard" is defined as any agent that can harm or damage the receiver and becomes a "risk" when the receiver is exposed to that hazard. We have compiled a comprehensive list of hazards to sea turtle health, which are not necessarily considered a risk for the species, but provide an exhaustive review of the published literature for future reference. Unpublished data was accessed through inter-discipline collaborators based around the world e.g. veterinarians and researchers from rehabilitation centres and universities (Fig 2).

For clarification, the disease hazards are divided into infectious and non-infectious and each of those further sub-divided to facilitate the risk assessment of each disease hazards.

To do this, Preferred Reporting Items for Systematic Reviews and Meta-analysis (PRISMA) was used to conduct a systemic review based on the recommendations by Foster *et al.* [27]. The global peer-reviewed literature databases such as Web of Science (1972–2016) and Scopus (1954–2016) were interrogated for literature relating to both infectious and non-infectious diseases of sea turtles. We therefore conducted two separate search strategies.

The search string for infectious diseases of sea turtles was:

(TS = (turtle*) AND TS = ( (infect* OR bacteri* OR vir* OR fung* OR parasit*) NEAR (green OR "Chelonia mydas" OR "C. mydas" OR loggerhead OR "Caretta caretta" OR "C. caretta" OR "kemp's ridley" OR "Lepidochelys kempii" OR "L. kempii" OR "olive ridley" OR "Lepidochelys olivacea" OR "L. olivacea" OR hawksbill OR "Eretmochelys imbricata" OR "E. imbricata" OR flatback OR "Natator depressus" OR "N. depressus" OR leatherback OR "Dermochelys coriacea"OR "D. coriacea") ))

Infectious means the entry, development/proliferation of a parasite in the body of a host, where it may or may not cause a disease: For this category, the following keywords were used: "infect*", "bacteri*", "vir*", "fung*" and "parasit*".

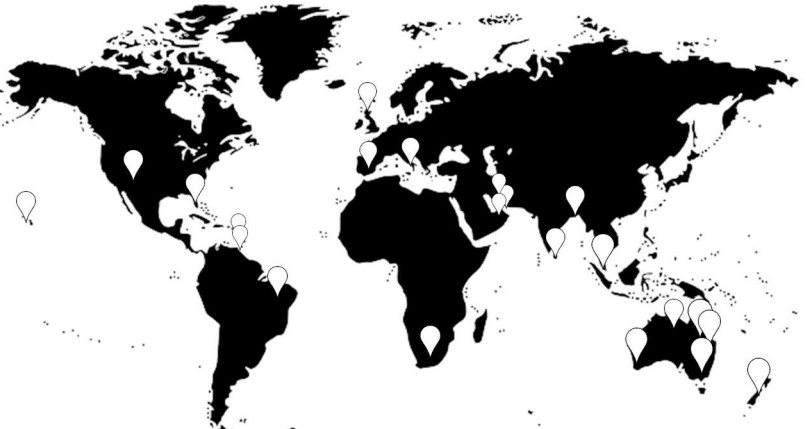

**Fig 2. Origin of contributors to the hazard identification and assessment of sea turtle diseases.** (The world map is an Image by Clker-Free-Vector-Images from Pixabay).

The search string for non-infectious diseases was:

(TS = (turtle*) AND TS = ( (disease* OR injur* OR nutrit* OR immun* OR disorder* OR syndrome* OR trauma* OR rehab* OR strand* OR mortal*) NEAR (green OR "Chelonia mydas" OR "C. mydas" OR loggerhead OR "Caretta caretta" OR "C. caretta" OR "kemp's ridley" OR "Lepidochelys kempii" OR "L. kempii" OR "olive ridley" OR "Lepidochelys olivacea" OR "L. olivacea" OR hawksbill OR "Eretmochelys imbricata" OR "E. imbricata" OR flatback OR "Natator depressus" OR "N. depressus" OR leatherback OR "Dermochelys coriacea"OR "D. coriacea")))

The non-infectious diseases refer to the diseases which cannot be transmitted between organisms and were extracted from the database using the following keywords: "disease*", "injur*", "nutrit*", "immun*", "disorder*", "syndrome*", "trauma*", "rehab*", "strand*" and "mortal".

The strings detailed above were then adapted to the Scopus database. The Web of Science and the SCOPUS search strategy yielded 568 and 627 publications, respectively, which were imported into Endnote X7® (Thomson Reuters®, 2017). The searches resulted in 1195 papers which were considered for further analysis (Fig 3).

After removing 573 duplicates, 622 papers were left. All publications (e.g. peer-reviewed manuscripts, conference proceedings, government reports and book chapters) examining the health status of sea turtles were chosen for further evaluation by a group of co-authors. These studies were screened based on titles and abstracts and 436 references were removed for the following reasons:

1. Repetitive results elaborating the same findings

2. Case reports with specific results from one species or one region

3. FP experiments, finding the disease in new locations, virological assays

4. Biochemistry of trace elements, drug pharmaco-kinetics and haematology of healthy and unhealthy sea turtles

Disagreements on study selection were resolved by consensus and discussion with other reviewers, if needed. 186 references were left for full-text assessment based on the literature search that was conducted in 2016.

The information was collated in the form of different sets of tables (S4-S8 Appendices in S1 File). In infectious health hazards category, pathogens were alphabetised and for each pathogen, if available, regions and species were reported along with the outcome of infection, transmissibility and possible correlation with climatic influence and anthropogenic events. In non-infectious diseases, five main groups (physical trauma, nutritional problems, environmental factors, anthropogenic problems and medical problems) were defined which were sub-categorised to more detailed health problems as seen in S8 Appendix in S1 File. For each health problem, regions and species were reported along with the aetiology, treatment, effects on population and mortality rate, if clear.

These tables were then presented in structured workshops (Turtle Health & Rehabilitation Workshop, September 2017, Townsville, Australia and Medicine Workshop at the International Sea Turtle Symposium 2018, Kobe, Japan). Six different groups of health hazards were formed for gram negative bacteria, gram positive bacteria, fungi, parasites, viruses and non-infectious diseases. Allocated experts in each team received printed documents for discussion and modification. The tables were then modified and expanded based on published and unpublished literature. The literature-based "hazard identification" (section 3.1) and "sea

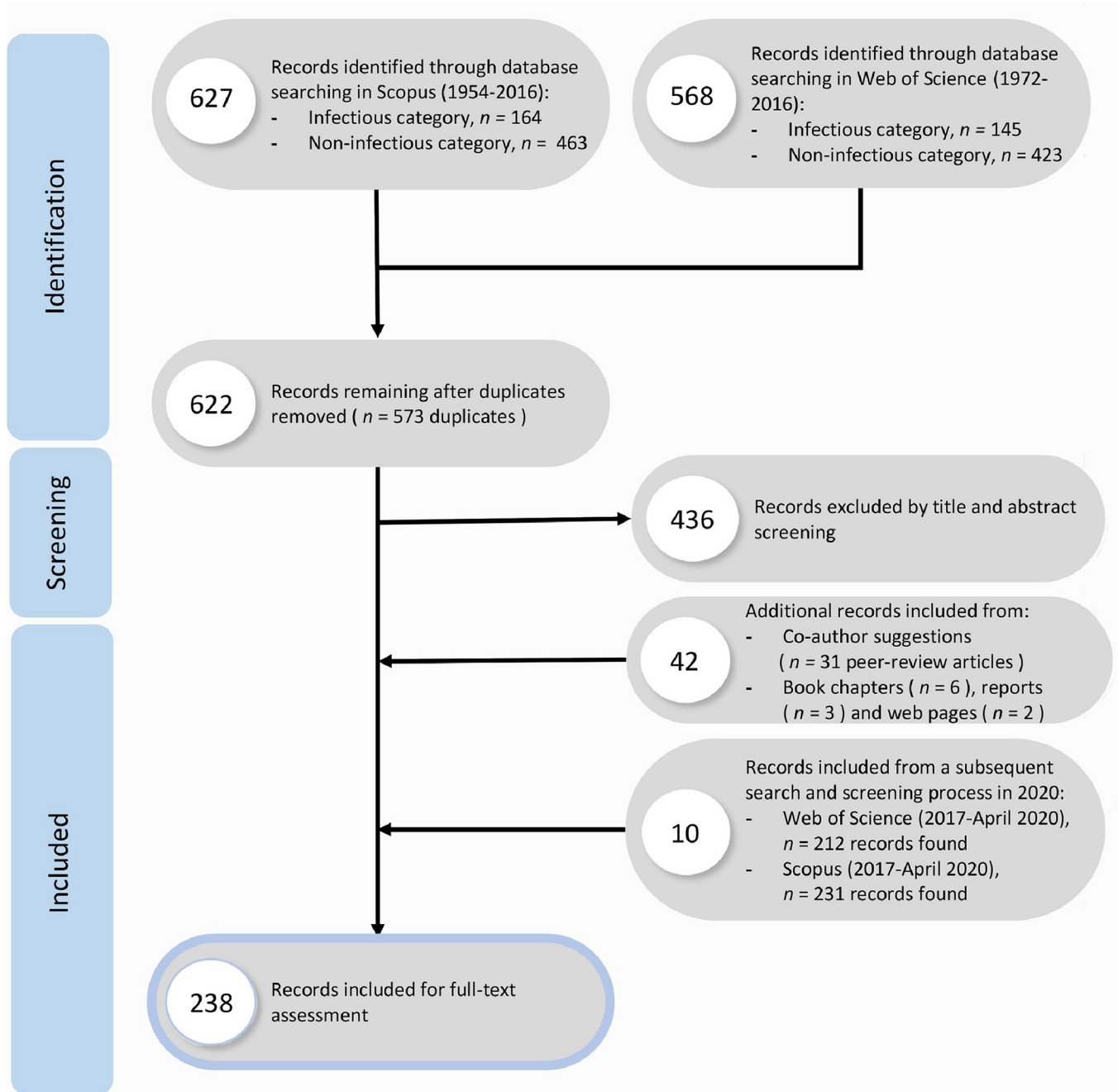

**Fig 3. The PRISMA flowchart displaying the selection procedure for writing the literate-based "hazard identification" (section 3.1), "sea turtle and One Health consideration in the literature" (section 3.3.1) and "S4-S8 Appendices in S1 File".**

turtle and One Health consideration in the literature" (section 3.3.1) were written by those workshop-participants that agreed on contributing to this manuscript.

To conduct a comprehensive DRA, unpublished manuscripts, gray literature and book chapters should be considered which are not accessible through advanced searches in databases such as WoS. To resolve this issue, additional records on diseases of sea turtles were extracted using Google search engine (2016–2019, Google LLC). This strategy resulted in 10

reports, news and book chapters. After reviewing the main draft, 15 peer-reviewed articles were also suggested by co-authors. Whenever possible, the authors or affiliated researchers/clinicians were contacted and invited to participate in this DRA. These experts were remotely interviewed and were asked to participate in adding unpublished information. Some were also invited to participate in the official workshop forums conducted on September 2017, Townsville, Australia and February 2018, Kobe, Japan.

A subsequent search was conducted on the 10 April 2020 to update the results. We found a total of 219 publications (duplicates removed), which resulted in 145 articles for full-text assessment. From these, 10 peer-reviewed publications were used to update the appendices with the most recent findings.

This resulted in a total of 221 records for the development of the DRA.

**2.2.1. Infectious hazards.** Infectious disease is among the top five reasons for terrestrial species extinction [28]. At present, the status of marine animals has not been assessed which highlights the need for further studies on infectious diseases in marine wildlife. In addition to directly threatening the biodiversity of free-living animals, wildlife diseases can also pose a threat to domestic animals and humans if wildlife act as a reservoir for pathogens [29].

The infectious hazards for sea turtles were categorised into four groups: bacteria, fungi, parasites and viruses. In each category, pathogens were listed alphabetically with available information summarised. Table 1 is an example of this information for a bacterial pathogen. As sea turtles are migratory species and inhabit different marine environments at different life stages [30], the geographical distribution of pathogens and host age were included, if known. Likewise, the presence of these pathogens in the wild and in captive populations were specified. The known infected or potential hosts were registered for each pathogen including related species, in order to address One Health considerations. Where possible, the correlation with climatic influence and/or anthropogenic events were also included to assess possible correlation.

**2.2.2. Non-infectious hazards.** Non-infectious diseases of sea turtles have been reported both in captivity and the wild [32], but little is known about the cause and extent of these diseases and their impact on the population [33]. In this study, a broad range of health problems were described to form a basis for discussing their possible effects on the population. The groupings were adapted from the method used by George (1997) [32] and consisted of four main groups, namely physical, nutritional, anthropogenic and medical problems. Table 2 shows an example of a physical problem and associated information. The regions where the hazards were reported in the literature are listed along with the species that were affected either in captivity or in the wild. For each health problem, the following information was collected (if available): clear description of aetiology, reports of mortality/ morbidity, effect on individuals/ populations and treatment availability.

**Table 1. Example of an infectious bacterial health hazard summary for "Lactococcus garviae".**

| Infectious health hazard | Region reported | Presence in sea turtle | | Outcome of infection* | Zoonotic/ transmissible to cohabiting animals | Correlation with climatic influence/ anthropogenic events | Key reference (s) |
|---|---|---|---|---|---|---|---|
| | | Captive populations | Wild populations | | | | |
| *Lactococcus garviae* | Tuscany, Italy | | Green (*Chelonia mydas*), Loggerhead (*Caretta caretta*) | Detected using PCR; | Present in fish, molluscs and crustaceans | Climate change may influence the threat levels associated with such exotic pathogens | [31] |
| | | | | No pathogenic studies carried out | Identified in a bacterial epidemic in aquatic invertebrates, such as the giant freshwater prawn | | |

*(lesion, clinical sign and/or disease) symptom in individuals; ease of spread, rate of spread; diagnostic test or treatment, if available.

**Table 2. Example of a non-infectious health hazard summary in the group of physical problems/injuries.**

| Non-infectious health hazard | Health Problem | Region Reported | Species Affected | | Aetiology, if clear; the effect on individuals, population, if known; treatment, if stated; mortality, morbidity if reported. | Key Reference(s) |
|---|---|---|---|---|---|---|
| | | | Captive Population | Wild Population | | |
| Physical Trauma | Injuries | Frequently reported | * | * | Due to predator bites, by-catch, boat strike or accidents | [34–36] |
| | | | | | May happen quite often and lead to infection, fractures and open fractures of a limb or of the shell, amputation of one or several limbs or minor wounds | |
| | | | | | Mortality may occur if the injury is traumatic | |
| | | | | | Appropriate modifications to vessel operation and configuration may reduce the threats | |
| | | | | | Aggressive males may bite females during mating | |
| | | | | | Captive turtles are prone to injuries in overcrowded facilities | |
| | | | | | Existence of rehabilitation centres in the area to surrender injured or caught turtles for healing period followed by releasing may help the population | |

* There is not enough information about the species or the region

## 2.3. Risk assessment

Two workshops involving experts with a broad range of expertise were convened to systematically execute the risk assessment step. The consultation process was conducted in a formal and structured manner following an established protocol for a DRA (see S2 Appendix in S1 File for workshop workbook and questionnaire) [18, 37]. Human ethics approval for this study was granted by James Cook University Human Ethics Committee, permit number H6834. The two international workshops were: 1) the Turtle Health & Rehabilitation Workshop, September 2017, Townsville, Australia, that was attended by 25 participants mainly from South Africa and the Australasia region and 2) the Medicine Workshop at the International Sea Turtle Symposium 2018, Kobe, Japan, where the 35 participants were from a broader range of regions and both hemispheres. The participants were veterinarians, microbiologists, members of the International Sea Turtle Society (ISTS) and IUCN Sea Turtle Specialist Group (MTSG) IUCN SSC Wildlife Health Specialists Group member and Widecast Coordinator (Saint Martin/Saint Barthelemy FWI) who are working on sea turtle research and conservation. Discussions among participants centred on the relevance, significance and prioritisation of infectious and non-infectious hazards.

The list of hazards, compiled in the review of the literature, were presented to the groups of specialists in sea turtle health. The "*Paired Ranking Tool*" was used to prioritise the top three hazards from each group according to a conservation, surveillance and research perspective [6, 18]. The paired ranking tool is a decision-making tool which is fully explained in Armstrong *et al.* [37] and Jakob-Hoff *et al.* [18]. The main goal of this technique is working out the relevant importance of the hazards. As mentioned by Jakob-Hoff *et al.* [18]: "This is a tool for a qualitative risk analysis that assists groups to rank hazards based on their collective judgement." The criteria used to compare the diseases were defined as: current knowledge of the pathogen in sea turtles, the likelihood of exposure/susceptibility, the pathogenic potential, the severity for populations and the correlation with climatic/anthropogenic events [6, 18].

## 2.4. One Health considerations and DRA

Both One Health and the DRA process share common goals, which are addressing complex health issues and aiming to reduce disease risks through multidisciplinary collaborations [38].

To address One Health considerations in this DRA, zoonotic pathogens of sea turtles and the possibility of disease transmission to/from sea turtles were documented. The information about socioeconomic consequences of conservation initiatives or the general benefits of interaction with sea turtles were also collected and reviewed.

Two sections were dedicated to One Health in the expert workshops: one addressed infectious disease transmission and the other explored opinions about the socioeconomic values of interaction with sea turtles and the contributions to conservation.

## 2.5. Risk management

Appropriate management interventions such as by-catch reduction, restrictions on commercial use and trade, and creation of protected habitats can allow recovery of a depleted population [39, 40]. This emphasises the importance of designing management with SMART (specific, measurable, achievable, realistic and time-based) goals [30]. Disease risk management is the process of risk evaluation and identifying the measures that can be applied to reduce or eliminate the risk posed to the population of concern [41]. To effectively reduce or eliminate the risks, the scale at which the management plans are evaluated and executed should be defined. Regional management units (RMUs) were developed for sea turtles to organise units of protection. These are functionally independent and provide a framework to evaluate conservation status and to address management challenges [42].

After defining the management unit, the risk management step suggests management options to reduce the risks that have been assessed and ranked in previous steps. These options are then evaluated according to their feasibility and effectiveness [18]. However, often this is not the case and as the options may not be ideal the best available under the existing circumstances will be selected.

Reducing the risk is not implemented under a "*single correct answer*" achieved from risk assessment, it is rather a step-by-step procedure that needs modification through communication and cross-governmental support as animals and their pathogens are not confined by political barriers but are distributed by topographic and ecological barriers [18, 41]. This is especially true for migratory animals such as sea turtles [30].

In most cases the risk assessment process is separate from the risk management implementation, merely because the scientists and veterinarians behind the risk assessment process are not policy or decision makers at government level [30]. However, the 'experts' are the ones that understand the biology and the ecological systems under consideration. Therefore, they are the best people to identify the range of risk management options. The policy makers should then have input into the feasibility evaluation of the options proposed. Hence, this is best done collaboratively rather than separately as the two groups need each other's perspectives to make the best decisions.

A scientifically based, clear DRA can help the decision makers to prioritise the actions to reduce the disease risk [18]. An understanding of the identified and assessed risk can facilitate practical and realistic interventions in the form of risk management of the most significant hazards [41].

At the international workshops, the DRA protocols were used to structure discussions around the current risk management, its difficulties and defects for the highest ranked hazards based on globally identified challenges for risk management initiatives. As executing risk management for a specific scenario and in a defined region is more realistic than a global disease risk management for sea turtle populations, the local workshops facilitated further discussions with appropriate representatives from the Australian government. The risk management workshop took place in February 2019 at James Cook University, Townsville, Australia and

aimed to identify possible pathways for local disease risk management. The attendees were provided with the DRA materials including the risk assessment results, a week prior to the meeting. The workshop workbook is provided in S3 Appendix in S1 File. The workshop was divided into two sections, the first part was discussing management options for previously assessed risks and the second part was brainstorming to define critical control points for a mock clutch translocation.

**2.5.1. Management options for previously assessed risks.** To follow the structure of the DRA, the management group selected two prioritised risks from the previous step "risk assessment". These two were the most relevant risks to Townsville local conditions which were also the highest ranked hazards in previous steps.

The first risk was "*Enterobacteriaceae and multi-resistant bacteria*" from the infectious hazard group. In 2017, researchers from James Cook University (JCU) reported that Enterobacterial isolates from rehabilitated green turtles were significantly multidrug resistant which has an implication for conservation actions and the general health in the Great Barrier Reef [43].

The second risk was "*macro-plastic pollution*" from the non-infectious hazards. It has been reported that green turtles inhabiting the Queensland coast, and generally in Australia, are exposed to macro- and microplastic pollution and unfortunately ingest plastic debris [44]. Management options were suggested for these two hazards by the attendees and effectiveness and feasibility were scored based on the discussions.

**2.5.2. Critical control points for a mock clutch translocation.** The translocation of animals for conservation purposes was the original and primary aim of establishing DRA [18]. The problem description, scope of the risk, goals of risk analysis and the source of information will vary for each individual scenario. Here, the hazard is confined to the regions that "*animals are sourced from*" and the destination that "*the animals are going to be introduced to*" [45]. The list of the hazards is mainly focused on the "*disease causing*" infectious and non-infectious agents. The risk assessment can be done through expert-involved discussions and scenario trees for a graphic representation of the specific translocation situation. Risk mitigation and contingency plans can be created with reference to the risk assessment. Finally, the stakeholders can plan for scientifically based, feasible and economic risk managements.

The checklist for conducting a wildlife translocation disease risk analysis [18, 46] was modified for a scenario of sea turtle clutch translocation and employed here as an example (See S1 Appendix in S1 File). Such procedures are relevant for hatching, captive rearing, rehabilitation and release of turtles, though individual and local considerations must be taken into account for each scenario. One example is head-start program which is designed to increase hatching rate by captive rearing the sea turtle hatchlings and releasing them to the ocean when they are assumed to have higher survivorship [47].

In the second part of the local workshop, risk management for a mock clutch translocation from an island to the mainland was assessed and the potential transmission pathways for infectious organisms were agreed on after discussing the modes of transportation of the eggs. The potential transmission pathways and the critical control points were then listed in a schematic representation on a whiteboard. Predation risk was also considered in the destination area and the potential hazards for a hatchery establishment were discussed.

## 3. Results

### 3.1. Hazard identification

Both infectious and non-infectious hazards were addressed and the complete list is available in S4-S8 Appendices in S1 File. Here we consider only those pathogens and diseases that are important in the context of sea turtle conservation and have left out a large number of

potential pathogens that would make the DRA unrealistic and unachievable [41]. Still, this study identified a comprehensive list of infectious and non-infectious hazards for consideration in this DRA.

**3.1.1. Infectious disease.** Previously undetected bacteria, viruses, parasites and fungi are frequently described in sea turtles and in new regions, but the health implications to sea turtles are not commonly addressed in the literature [15]. In many cases, this makes it difficult to determine how high-risk some hazards are, highlighting the need for expert opinion. An exhaustive literature search identified the following information on possible hazards of interest to this DRA.

*3.1.1.1. Bacteria.* Most bacterial species in sea turtles are opportunistic pathogens and have been reported as natural flora in fish, crustaceans and other marine animals [7]. In early studies, bacterial pathogens formed the longest list of infectious hazards for sea turtles contributing to disease in captive, farmed and free-living sea turtles in many parts of the world [48–50]. The list of bacterial pathogens has grown (see S5 Appendix in S1 File) in terms of diversity but not necessarily the prevalence and the effect on the population.

*Vibrio spp. Pseudomonas spp.*, *Enterococcus spp.*, *Aeromonas*, *Cytophaga. freundii*, *Escherichia. coli*, *Edwarsiella spp.*, *Proteus spp.*, *Lactococcus garviae*, and *Providencia* have been recorded in sick sea turtles as either potential primary pathogens or opportunistic bacteria [31, 51]. *Vibrio spp.* are the most frequently studied bacterial isolates in sea turtles (especially *Vibrio alginolyticus*) and are repeatedly isolated from skin lesions, digestive organs and respiratory tract associated with ulcerative stomatitis, obstructive rhinitis, and pneumonia along with *Aeromonas hydrophila*, *Pseudomonas fluorescens*, *Flavobacterium spp.*, and *Bacillus spp.* [48, 49, 52]. Infection with these bacteria can also cause mortality in captive-reared and/or wild juvenile green and loggerhead turtles [48, 49].

Bacteria isolated in clinically healthy and wild-living turtles near urbanised areas show high levels of multidrug-resistance, indicating an accumulation of resistance in marine bacteria caused by exposure to anthropogenic factors. Of particular concern are the *Enterobacteriaceae* that are of One Health importance as potential zoonotic pathogens [43].

*3.1.1.2. Fungi.* Fungal pathogens of sea turtles are usually opportunistic saprophytes causing infection under favorable circumstances [53]. Sea turtles in captivity or rehabilitation centres are prone to mycotic infections possibly due to other underlying health issues or immunosuppressive conditions [7]. *Fusarium* species have been isolated from cutaneous abscesses [54], cutaneous or pneumonic lesions and bronchopneumonia [55]. *Fusarium solani* is the most frequently identified fungus in sea turtle mycotic diseases, and is normally isolated and referred to as a 'species complex' including more than 60 phylogenetic species [55]. *Fusarium* is widely distributed in soil and waste; it tends to enter the body through lesions, causing mycosis in humans and animals [55, 56]. *Fusarium* infections are a common pathological finding in sea turtle eggs; *Fusarium oxysporum*, *F. solani* and *Pseudallescheria boydii* were isolated from failed eggs found in eastern Australian loggerhead, green, hawksbill (*Eretmochelys imbricata*) and flatback (*Natator depressus*) nests [57]. *Fusarium falciforme* and *Fusarium keratoplasticum* were believed to reduce the hatching success to 10% per infected clutch [55]. Environmental stressors such as inundation (flooding of nest) and oxygen depletion seem to enhance the incidence of fungal infection and mortality of embryos [55]. However, Phillott and Parmenter [58] determined that the fitness of the hatched green turtles was not affected by fungal colonisation of the nest. Sporadic opportunistic fungal infections are reported in sea turtles. These fungi are not true pathogens of reptiles and are usually not associated with systemic infection or mortality unless the immune system is compromised [59].

*3.1.1.3. Parasites.* A variety of parasites infect sea turtles, primarily digenetic trematodes and nematodes [60]. Different factors influence the extent of damage a parasite may cause,

such as the species of parasite and the general fitness of the host, habitat and availability of intermediate host [60, 61].

The gastrointestinal flukes (digeneans of the family Pronocephalidae) and cardiovascular flukes (Spirorchidae) are the most prevalent trematodes in sea turtles [60, 62]. Gastrointestinal flukes are widely distributed throughout the gastrointestinal tract without any apparent ill effect. Cardiovascular flukes, on the other hand, cause pathological effects in the circulatory system and multiple internal organs [60]. The first definitive life cycle for a species of blood flukes in sea turtles was recently described with vermetid snails as the intermediate hosts for *Amphiorchis sp* [63].

In the nematode group, Anisakidae and Kathlanidae have been reported to infect sea turtles and are mainly found in the gastrointestinal tract of loggerhead turtles [61, 64]. In Australia, the coccidian parasite *Caryospora cheloniae* and Spirorchiids are reported to be the parasites of highest concern as they are associated with disease and high mortality rates under certain conditions [65]. Of the two parasites, Spirorchiids is reported to be more common and widespread [66].

Sea turtles are the definitive host for some of these parasites, but how host-specific or harmful these parasites are to the host is not known. *Lophotaspis valley*, *Learedius learedi* and *Styphlotrema solitaria* are some species-specific trematodes in sea turtles, while *Plesiochorus cymformis*, *Rhytidodes gelatinosus*, *Enodiotrema carettae* and *Pleurogonius trigonocephalus* have a wider host range [60].

*3.1.1.4. Viruses.* Reptile virology is a relatively new field [67]; however, increased awareness and advances in molecular technology will undoubtedly bring about an increase in the knowledge and identification of new species [68]. The link between the presence of herpesvirus or ranavirus and clinical disease in chelonians are well established, whereas the link between disease and causative pathogen is still being explored for other viruses [67]. To date, members of *Herpesviridae* are the only causative agents of viral diseases investigated in sea turtles. The presence of other viruses in sea turtles are sporadically reported: with one published report for each of tornovirus, retrovirus and betanodavirus [31, 69–71] and two reports of papillomaviruses [69, 72].

Herpesviruses cause severe diseases in chelonians, especially in animals in stressful situations with associated lower immune function [73]. Gray-patch disease (GPD), lung-eye-trachea disease (LETD) and FP are herpesvirus-associated diseases frequently described in sea turtles [26, 74–76].

GPD was reported in captive reared green turtles (less than year old) causing gray skin lesions. Overcrowded hatcheries and higher water temperatures appears to worsen the symptoms [77]. LETD, another disease of green turtles (over one year old) was first described from turtles in captivity and then found in free ranging green turtles [78–80].

Fibropapillomatosis is a neoplastic disease affecting all species of sea turtles [81–84]. Tumour growth can be both external and internal, with juvenile turtles appearing to be most susceptible. Moreover, infected turtles are vulnerable to secondary infections and opportunistic pathogens due to immunosuppression [82, 84]. Environmental factors may contribute to the expression and the severity of the disease [82, 85, 86]. The disease was first reported in an aquarium in New York [87], but is now reported globally in tropical waters [51, 84, 88–90].

**3.1.2. Non-infectious diseases.** Turtles are affected by a variety of non-infectious diseases occurring either as a direct result of natural or man-made threats [32], or they may act as multifactorial influences on disease outcome. In some cases, it is not easy to determine if clinical signs are caused by an infectious or non-infectious agent. Infection with coccidia can elicit neurological diseases, but neurological symptoms can also be caused by head injury or natural causes such as toxins and algal bloom [91].

Serious alterations in the balance between the environment, the host and the pathogens can trigger or spread disease in a population [18, 92, 93]. For example, loss of seagrass habitat due to human disturbances or severe weather events can influence water quality and lead to immunosuppression due to starvation [94, 95]. Anthropogenic effects such as habitat degradation, coastal light disturbance, pollution, and by-catch are known threats posed to sea turtles and are ranked highest in terms of adverse effects they may have for sea turtle populations [96, 97]. The flow-on effect of habitat disturbance for turtles are likely to facilitate the emergence of infectious diseases at increasing incidences and exacerbate the risk of local population extirpation [94].

*3.1.2.1. Trauma and injuries.* Traumatic injuries are a major cause for stranding and may be caused by a range of factors from boat strikes and entanglement to shark bite or mating injuries [49, 97, 98].

Air breathing marine species are at risk of vessel collision and sea turtles are no exception [12]. The data clearly indicates that the vessel strike injuries occur in both protected marine environments and remote areas [99].

As examples, during 2000–2014 in Florida, between 1,326 and 4,334 sea turtles mostly green and loggerhead were killed after a boat strike incident [100]. In the French Caribbean, boat strikes appear to occur at a higher rate in the past few years (personal interview with the veterinarian in Saint Barthelemy/Saint Martin FWI). The incidences are lethal in the majority of cases with a very low survival rate (1/10 sea turtles survived after intensive veterinary care at Saint Barthelemy/Saint Martin FWI in 2019). In Galapagos Marine Reserve, incidences mostly happen in commercial tourism areas and there is no data on survival rates [99].

To diminish this threat different laws and regulations have been imposed [101] and to reduce the deadly impacts several conservation approaches were introduced such as educating vessel operators on how to spot sea turtles, establishing go-slow zones or no-entry areas and increasing the use of propulsion systems to avoid exposed propellers [12, 100].

Mortality caused by fishing gear is also troubling sea turtle populations around the world. The data from a 5-year study (2013–2017) in the Mid-Atlantic and Northeast has reported the alarming mortality rates caused by different fishing gears: trawl, 48%; gillnet, 73%; dredge, 40%; vertical, 55% and fish trap, 57% [102]. As explained in Canary Islands, Spain, fishing is associated with net entanglement and hook and monofilament line ingestion which lead to death [52]. Implementing changes in fishing strategies, operations and technologies have shown to reduce the mortality rates associated with fishing worldwide [103, 104]. An example is using turtle exclusion devices (TED) to be able to release sea turtles from trawls [103].

In addition to direct lethal effects on individual turtles, open wounds are a portal of entry for pathogenic microorganisms into the turtle [97]. Perforating fishing hooks, plastics and fish spines can cause injuries in the gastrointestinal tract and respiratory system after ingestion [15, 97].

Decompression sickness was also recently diagnosed in loggerhead turtles captured in trawl and gill nets in the Mediterranean Sea [105].

*3.1.2.2. Debilitated Turtle Syndrome (DTS) and cold stunning.* Debilitated Turtle Syndrome (DTS) is used to describe the condition of a turtle with several of the following symptoms: emaciation, lethargy, hypoglycaemia, anaemia, and heavy coverage with epibiota [106]. Secondary infections are common in turtles with DTS, and turtles may be immunosuppressed [107]. A wide range of morphometric and metabolic variables is documented for chronically debilitated loggerhead turtles in the southeastern United States [106]. The main cause of DTS is not clear but cold stunning in some cases is an initial trigger [108, 109]. Occasionally, large numbers of strandings are reported due to cold stunning based on the personal interviews with rehabilitation centres from Dubai, UAE; Kish Island, Iran; New York, USA; Lampedusa,

Italy in 2017. Epibiota can increase rapidly in numbers when turtles are floating or immobilised and due to the invasive nature of some species of these epibionts, they can be detrimental to health. A high load of epibionts can lead to erosion in the carapace and plastron creating a portal of entry for secondary invaders [15].

*3.1.2.3. Gastrointestinal disorders.* Gastrointestinal disorders are one of the main concerns for sea turtles in rehabilitation centres [43]. Gastrointestinal obstruction by debris such as plastic are a clear risk for turtles [110, 111]. Macroplastic ingestion is ranked at the same level as other anthropogenic pressures such as by-catch [112]. The risk of plastic ingestion has been documented for decades and has affected different age groups of sea turtles [113–115]. As a result, in post hatchlings and juvenile turtles, gastrointestinal impaction or injury directly affect the health and reduced nutrient intake or abnormal buoyancy indirectly impair the fitness [113]. Generally, the risk of macroplastic ingestion is documented more than microplastics and in some cases is reported to have higher impacts. Micronising plastic can accumulate in coastal sediments and in ocean beds but the consequences for sea turtles is still unclear [112].

Gut impaction and faecoliths are also observed in stranded sea turtles with no obvious or physical cause [15]. Climatic events may alter the foraging grounds for turtles and thereby affect their nutritional choices [116], but physical trauma, high parasitic load or chronic diseases can lead to loss of appetite, nutritional deficiencies and cachexia [32]. Nutritional disorders can in turn affect the hepatobiliary system [15].

*3.1.2.4. Diseases caused by chemical and organic pollutants.* Pollution can cause immune suppression and thereby increase vulnerability to pathogens [92]. Organic agricultural waste can elevate the nutrient level in the ocean and stimulate harmful algal and cyanobacterial blooms which can directly or indirectly harm turtles or exacerbate the effects of other diseases such as FP [117–119]. In addition, long living animals, such as sea turtles, face the risk of accumulating these pollutants in their tissues over time and as a result the impact of toxicity may intensify [97].

Chemical debris and organic pollutants can block the gastrointestinal tract and cause different problems such as accumulation of intestinal gas, local ulcerations, interference with metabolism and immune function and intoxication [110, 111, 119]. Plastic is an example of an accumulating pollutant and sea turtles tend to ingest plastic debris [120] which may block the gastrointestinal tract, accumulate intestinal gas, cause local ulcerations and interfere with metabolism [110, 115]. Gastrointestinal obstruction may lead to chronic debilitation and eventually death [121]. Cases of secondary infection and mortality are frequently reported due to plastic ingestion [114, 115].

Anthropogenic non-infectious diseases are the biggest challenge to sea turtle conservation [15, 122].

**3.1.3. Global warming and diseases.**   Global temperature is increasing and may directly influence species or the pattern of their lives [123, 124]. Migratory species are more vulnerable to temperature change, especially if they are forced to undertake large distance migration for breeding or changing seasonal habitats [123]. Climate change can also alter pathogen survival, transmission patterns, ecological balances, vector and host susceptibility [124, 125]. Such adverse impacts have been seen in FP prevalence and the development of clinical disease [76].

Sea turtles exhibit temperature dependent sex determination and global warming is proved to exacerbate female biased offspring in these species. The sex ratio of 90% females is reported in some nests which is reducing the chance of reproduction in a population [126]. At birth, hurricanes and other storm events inundate the nests and lead to hatching failure [86, 127]. After birth, climate change can threaten their habitat when severe weather events disturb the sea grass meadow, lead to coral bleaching and influence the water quality. Climate change is

adversely affecting sea turtle populations, however to directly link global warming and the health of sea turtles and to describe it as a causative agent for diseases, further investigations are required [95].

In 2010, two independent "Expert Opinion Surveys" were done to rank the threats for sea turtles. The results were inconsistent; one group ranked the global climate change and pathogens the least important threat for sea turtles [128] while the other group gave the highest rank to global warming [122]. Experts' opinion around these impacts is filled with uncertainty, which is reflected in the very different ranking of climate change in the two publications. The authors believed the differences in ranking came from the recent identification or emergence of these hazards which also emphasise the need for further investigations on this issue [128].

## 3.2. Risk assessment

To assess the disease hazards outlined above using expert opinion, group and forum discussions were facilitated and encouraged in the workshops. The discussion sessions, which formed the basis for the rankings, were an opportunity for the participants to explain their personal experiences with disease encounters and to improve the general knowledge of the participants about regional differences in disease manifestation. One point that was repeatedly mentioned was the "*quality of information available*" and how this affected the ranking. It is worth mentioning that this is a feature of all wildlife DRAs and the process enables information gaps to be identified and the level of uncertainty made explicit. Such level of confidence by experts is referred to in Pacioni's ranking criteria as "*levels of knowledge*" [6].

The top three hazards from each group of infectious and non-infectious hazards were ranked according to a conservation, surveillance and research perspective (Table 3).

Although the outcome from the two workshops are very similar, there were a few differences, which could reflect the broader geographical origins of participants in Workshop B compared with Workshop A. In Workshop B the experts working on parasites ranked the hazards based on overarching classification, while participants in Workshop A gave species names to the parasites. In both workshops, Spirorchiids were considered important due to their widespread presence and potential virulence. Ozobranchid leeches were also mentioned by both groups due to their possible role in FP transmission. Viral pathogens were considered to be data-deficient by participants in both workshops, but both groups listed herpesvirus and papillomavirus as the highest-ranking pathogens.

Antibacterial resistance and the associated public health concern were also consistently mentioned in the two workshops for the bacterial category. In Workshop A, the participants chose to focus on Gram negative bacteria only. *Fusarium* and *Cladosporium* spp. were selected by both groups as the most important fungal pathogens, mainly for eggs on nesting beaches and hatchlings in captive situations. Climate change and anthropogenic impacts scored highest in non-infectious health hazards in both workshops and there was consensus, that anthropogenic influences on turtle health need the highest attention of all groups, both in terms of research and conservation management.

## 3.3. One Health and DRA

**3.3.1. Sea turtle and One Health consideration in the literature.**   Sea turtles mostly encounter humans during harvest, on nesting beaches and in rehabilitation centres. Fig 4 shows the main sources of interaction between humans, sea turtles and the environment. These interactions can positively or negatively impact sea turtles and those trying to protect them.

**Table 3. The three highest ranked hazards of each infectious and non-infectious groups as determined by panels of experts in two international workshops.** A) Turtle Health & Rehabilitation Workshop, September 2017, Townsville, Australia, B) Medicine workshop at the International Sea Turtle Symposium 2018, Kobe, Japan.

| A. Turtle Health & Rehabilitation Workshop, September 2017, Townsville, Australia | | |
|---|---|---|
| **Hazard** | | **Notes** |
| **Infectious health hazards** | | |
| Parasite | *. Spirochiidae* | Widespread, virulent and prevalent |
| | *Caryospora cheloniae* | Virulent and episodic |
| | *Ozobranchus branchiatus* | Possible vector for FP associated herpesvirus |
| Virus | Chelonid alphaherpesvirus 5 (ChHV5) | Associated with fibropapillomatosis: reported in all species, can cause debilitating syndrome and be life threatening |
| | *Chelonia mydas* papillomavirus (CmPV-1) and *Caretta caretta* papillomavirus (CcPV-1): | Skin lesions, data deficient |
| Gram negative bacteria | *Vibrio spp.* | Associated with ulcerative dermatitis, mortality reported; associated with hatching failure; possibly zoonotic for turtle meat and egg consumers. |
| | *Pseudomonas spp.* | Ulcerative stomatitis and dermatitis along with *Vibrio alginolyticus*; associated with hatching failure; possibly zoonotic for meat and egg consumers |
| | *Escherichia coli* | Antibiotic resistant; opportunistic pathogen; zoonotic |
| Gram positive bacteria | Unfortunately there was not enough time to go through this list. | |
| Fungal infection | *Fusarium spp.* (mostly *Fusarium solani*) | Contributing to hatching failure, pneumonia, necrotic skin lesions mostly in captivity; potentially zoonotic. |
| | *Aspergillus spp.* | Hatching failure, mycotic infections in hatchlings; mycotic infections in captivity |
| | *Cladosporium spp.* | Hatching failure, infections in captivity |
| **Non-infectious health hazards** | | |
| | Anthropogenic: Habitat degradation | Malnutrition, by-catch and accidents |
| | Environmental: Climate change | Malnutrition, fibropapillomatosis and cold stunning or Debilitated Turtle Syndrome |
| | Anthropogenic: Pollution/plastic | Entanglement, external and internal injuries, debris ingestion and neurological diseases |
| **B. Medicine workshop at the International Sea Turtle Symposium 2018, Kobe, Japan** | | |
| **Hazard** | | **Notes** |
| **Infectious health hazards** | | |
| Parasite | *Spirochiidae* | Geographical wide distribution, various species, high prevalence, different effect in different life stages, adult, juvenile, eggs, severe lesions, causes stranding and mortality. |
| | *Annelids* | Wide geographical distribution, various species, Loggerhead, Olive Ridley and Green turtles are affected, cutaneous ulcerations, *Ozobranchus* possible vector for FP |
| | *Arthropods* | Needs justification, worse in some regions, correlated to hatching failure and egg damage, causing mortality, regional reports |
| Virus | Herpesvirus | Tumours have been reported globally, ChHV5 is reported in clinically healthy turtles |
| | Papillomavirus | Only a few reports so far, not fully understood |
| Bacteria | *Methicillin-resistant Staphylococcus aureus (MRSA), E. coli and E. margonella* | Multi-resistant strains, public health concern |
| | *Streptococcus iniae, Salmonella typhimurium, E. coli.* | Pathogenic and zoonotic |
| | *Pseudomonas spp. Klebsiella* | Mass mortalities, regional |
| Fungal infection | *Fusarium solani* | Problem for captive rearing, eggs and hatchling |
| | *Penicillium spp.* | Recorded in several areas, multi species infection recorded, different stages of life can be affected |
| | *Cladosporium spp.* | Recorded in several areas, may affect several life stages |
| **Non-infectious health hazards** | | |
| | Anthropogenic | Human interactions are increasing, plastic ingestions are increasing |
| | Environmental | Climate change effects and also cold stunning |
| | Medical | Aftermath of anthropogenic and environmental incidences |

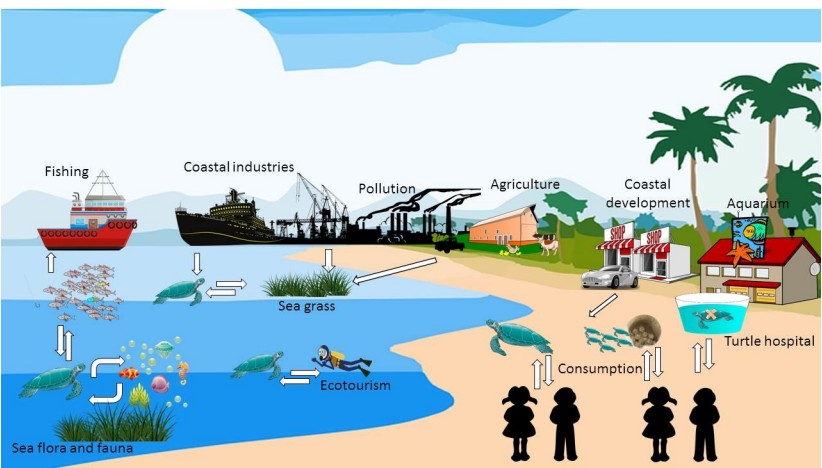

**Fig 4. The schematic interactions between sea turtles, humans, co-habiting animals and the environment (the vectors, characters and icons in this figure were downloaded from the public domain https://pixabay.com/ and were modified according to its licence).**

*3.3.1.1. Zoonosis.* As an example of zoonotic infections, vibriosis in humans may develop due to consumption of contaminated meat and eggs [129]. Field workers should consider disinfecting any wound received while handling sea turtles as there is the risk of infection with *Mycobacterium*, *Salmonella*, *Vibrio*, and *Chlamydia* species due to contact with infected animals [75]. There are also reports of fish pathogens in sea turtle which are of concern to aquaculture and the sea food industry [31].

*Fusarium solani* can infect egg clutches, and high mortality rates are reported due to infection with this species of fungi. Being zoonotic, this pathogen poses a threat to the person handling the infected eggs as well. Such activities may take place while eggs are collected for consumption or in the hatcheries or on nesting beaches when the nests are cleaned out after the eggs are hatched. Dead/decomposing embryos are sources of nutrients for bacterial and/or fungal growth [130].

Toxins may not necessarily be categorised as zoonotic agents but can have ill effects on humans. Sea turtles are exposed to toxins of either anthropogenic or natural origins, which may accumulate in their tissues and cause problems for meat and egg consumers [129]. There are multiple reports of death, mass poisoning or sickness in a community after feasting on turtle meat [131–133]. The condition is termed *chelonitoxication* and appears to be caused by the consumption of particular sea turtle species (green, hawksbill and leatherback turtles). Children are more prone to intoxication and its lethal effects [131–133].

Humans can be the source of infection for sea turtles too. Examples of *Salmonella* and *Vibrio alginolyticus* transmission in captivity have been reported several times [7, 50–52, 134, 135]. Humans are also posing an indirect threat to sea turtle health, via habitat destruction, distribution of pollutants, plastic and toxins [22, 23].

*3.3.1.2. Cultural significance and sustainable conservation measures.* Sea turtles are of great cultural value for indigenous communities [136]. Humans and their environment co-evolve, and local culture and traditions reflects this relationship. The legally recognised rights of indigenous communities to interact with sea turtles in line with their traditions is the foundation for a community-based conservation management where alternatives to hunting is introduced in consultation with the local communities (e.g. Caribbean Coast of Nicaragua) [137]. Such

policies reduce the fear of arrest or reprisals while participating in local customs [137] which in turn enhances the feeling of control over their lives and improves community health.

Market-based solutions towards conservation and providing alternatives for consumption of sea turtle products have been successful in several projects such as the Tartarugas Marinhas (TAMAR) project sites in Brazil, and at Tortugeuro, Costa Rica [137]. At these locations the hunting has decreased, while ecotourism-based activities have been organised for local communities. Other non-governmental organisations (NGOs) have also formed and evolved in various regions of the world to promote conservation with the help of local communities. One such example is New Idea in Hormozhgan, Iran (in Persian: moassese ide no doostdare hormozgan) which was successful in eliminating egg harvest for overseas markets. The turtle nesting site is now an ecotourism destination with a financial return for the local community (personal interview with Maryam Eghbali the co-founder, 2017). A pro-environment establishment, Grupo Tortuguero, was formed in the Pacific Ocean in response to poaching and retaining the turtles after accidental catch by fishermen. The establishment is active in terms of education, funding and empowerment in response to loss of sea turtles, especially loggerheads [138].

Governments can also work in partnership with traditional owners to manage and conserve species. In Australia, Traditional Owner groups can develop an agreement on how they will manage traditional activities on their sea country. This agreement, or Traditional Use of Marine Resources Agreement (TUMRA), details how Traditional Owner groups wish to manage their take of natural resources (including protected species). This extends to their role in compliance and in monitoring the condition of plants and animals, and human activities, in the Great Barrier Reef Marine Park. Once developed, a TUMRA can then be accredited by the state and federal governments [139].

When sea turtle conservation does not limit people's ability to interact with sea turtles, it can have a positive impact on communities. Moreover, such conservation efforts can impact on the entire social-ecological system in which both turtles and humans are embedded [137]. Sea turtle conservation plans must therefore articulate with diverse cultural, political and socioeconomic needs [140]. This poses a challenge to management policies and raises important questions about the purpose of research and conservation endeavours [30]. As an example, in a recent publication by Barrios-Garrido *et al.* [140], the conflicts related to sea turtle conservation programs in the Caribbean basin were identified. Dissimilar conservation objectives between local communities, non-governmental and governmental organisations were identified, along with lack of resources such as trained individuals for monitoring and enforcement roles, and scarce funding [140]. The suggested solutions for these conflicts were rationalising the problem and promoting a mutual agreement based on common beliefs. Such multiscale solutions would be achievable by co-management through bottom up (community based) actions and top-down changes (government policy) [140].

**3.3.2. Sea turtle health and One Health according to expert opinion.** Several experts presented their experiences or One Health related case studies to share their specific challenges and the way they address these issues. The One Health discussions in the workshop centered around the transmission of pathogens between sea turtles in the wild and captivity, non-infectious disease transmission between humans and turtles and the cultural/socioeconomic aspects of sea turtle conservation. Ultimately, the expert opinion on disease transmission was consistent with the literature (Table 4).

It was agreed that *Fusarium solani* is the main concern for turtles in captivity and a threat to egg and meat consumers. In the non-infectious category, chelonitoxication and the mass poisoning it causes was considered of great importance. The pathogen transmission routes need further research to better understand the mechanisms at play. New hatcheries are being

established in some areas to take economic advantage of tourism, without following strict hygiene protocols (e.g. wearing gloves while handling the eggs and hatchlings or relocating nests) and the biological needs for the eggs to hatch (e.g. the correct temperature, adequate depth of the nest, how to handle the eggs). Another example is hand-feeding of sea turtles in the wild. Local guides or fishermen in some areas of the Canary Islands and Bahamas were reported to feed the turtles in the wild and there is concern about the health and behavior of the turtle population after being habituated to people.

In the workshop, the discussion about the cultural dimensions of interacting with sea turtles or the importance for indigenous groups concluded that there was a lack of knowledge in this field among the participants which highlighted the need for more social science studies. However, the social scientists present in the workshop shared their experience in this field. Social science experts work directly with the communities that interact with sea turtles. According to their experience, sea turtle conservation brings the communities together and gives them a common cause and sense of belonging to the environment.

These results highlight the multiple and intersecting One Health considerations in sea turtle conservation which should be considered in effective sea turtle management plans.

### 3.4. Risk management

**3.4.1 International workshops.** The current global management for the highest ranked hazards in risk assessment step are reported (Table 5) along with the difficulties and defects for each strategy. One Health considerations are also reported however, data deficiency about zoonosis and biotoxicity limit the ability to provide recommendations to egg and meat consumers. Several management options were suggested for socioeconomic aspects of interacting with sea turtles, however this list provided here is not exhaustive.

**3.4.2. Local workshop.** In the local risk management workshop, the overarching concern was inadequate communication between different sectors working on sea turtle surveillance and conservation. The attendees referred to the lack of comparable and accessible data for researchers, conservationists and government sections. The reason behind "data protection" or limited information sharing can be confidentiality, or variations in legislation for different organisations collecting such information. Nonetheless, such data protection impacts on the success of conservation initiatives.

*3.4.2.1. Management options for previously assessed risks.* The management options to reduce the risk of 1) macroplastic pollution and 2) *Enterobacteriaceae* and multi-resistant bacteria were ranked based on effectiveness and feasibility on a scale of 10, with 1 being the lowest and 10 being the highest. The results of these rankings are summarised in Table 6. It is important to note that in almost all cases a final decision on option selection was beyond the scope of the group.

The management scale for macroplastic pollution can be as small as a school or as big as the Queensland state. The group suggested that it should be divided to two categories: 1) eliminating the impacts of the macroplastic that has already been released into the environment and 2) to reduce further input. For the first category, promotion of beach clean-up initiatives and rubbish collection; installing storm drain filters, which requires local and external donors and long-term monitoring; promotion of funding for large scale ocean clean-up projects. For the first category, the options to reduce the production and/or input included, but were not limited to: education and awareness to reduce littering and use of disposable plastics; research on providing affordable biodegradable items and; government policies targeted to eliminate the use of single use plastics such as that initiated in Queensland in 2018 [142].

**Table 4. One Health consideration in disease risk analysis workshop.** A) Transmission of pathogens between sea turtles in the wild and in captivity. B) Non-infectious disease transmission between human and sea turtle. C) Cultural values of sea turtles and socioeconomic aspects of sea turtle conservation.

**A. Transmission of pathogens between sea turtles in the wild and in captivity**

| Pathogens | Main zoonotic pathogens of concern from turtles to humans | Pathogens being naturally transferred from humans to sea turtles | Main problematic pathogen in captivity for turtles | Pathogens to be considered as a risk for aquaculture and fisheries |
|---|---|---|---|---|
| Bacteria | *Salmonella* | Very unlikely | Opportunistic bacteria | Data deficient |
| | *Vibrio spp.* | | | |
| | *Pseudomonas spp.* | | | |
| | *Escherichia coli* | | | |
| Fungi | Fusarium (*especially F. solani*) | Data deficient | Fusarium (*esp. F. solani*) | *Trichophytea spp.* |
| | Aspergillus | | | |
| Parasites | Not a concern to date | Not a concern to date | Cryospora | Data deficient |
| Viruses | Not a concern to date | Not a concern to date | Herpesvirus | Herpesvirus |

**B. Non-infectious disease transmission between humans and sea turtles**

| Human to turtles | Turtles to humans |
|---|---|
| Biotoxin pollution | Toxins in egg and meat |
| Plastic pollution | |
| Boat strike, by-catch | |

**C. Cultural values of sea turtles and socioeconomic aspects of sea turtle conservation.**

Cultural dimensions of interacting with sea turtles have recently been brought to the attention of conservationists:

• Rescue plans are rewarding for volunteers, rangers and people who are involved.
• In the Caribbean, the conservationists' goal is to interact with the locals and to allow traditional harvest in sustainable manners. However, in some island such as French Caribbean Sea turtles are fully protected for nearly 30 years and the harvest is absolutely prohibited.
• In the Maldives, sea turtles can be kept as pets". The consulted expert emphasised the special bond between the turtles and humans.
• In the French Mediterranean, the aim is to involve fishermen in conservation initiatives to reduce the threat of by-catch.
• In Australia, sea turtles are significant elements of indigenous culture and any conservation plans is considering their traditional expertise

Socioeconomic advantage of sea turtle conservation which need more attention:

• Tourism value of healthy turtle population has not been evaluated
• Turtle watching tours are alternatives for fishing and has been successfully established in some regions.
• Job generation through alternative projects may reduce poaching but needs more research.
• Outreach opportunities to groups that are interested but not normally involved in sea turtle conservation
• Sea turtles are charismatic species and on third highest ranked animal for conservation initiatives.
Turtles are indicators of environmental health, but the association between their health and the environmental health need more research and potentially funding.

For reducing the risk of *Enterobacteriaceae* and multi-drug resistant bacteria, the experts reiterated that a preventive solution which promotes education and awareness would be useful. This would include promoting personal and protective equipment when working with sea turtles. The experts also suggested that reducing the prescription and consumption of antibiotics would help manage this risk. The post-release management options included extracting the antibiotics from sewage water and promote funding for research into solutions for this procedure. The feasibility and effectiveness of these options were scored in Table 6.

*3.4.2.2. Critical control points for a mock clutch translocation.* The clutch translocation scenario and critical control point allocated by experts in the local management workshop are shown in Fig 5.

Time management and temperature control were suggested to be critical for transporting the eggs. Personal protective equipment (PPE) and hygiene were proposed as the effective and feasible options to avoid the risk of pathogen transmission. Screening the relocation site for potential pathogens was suggested, however, the feasibility was ranked low. Nest protection and monitoring to reduce the risk of predation was critical to justify the time and cost spent for translocation. The group suggested development of protocols and surveillances for hatchery establishment.

**Table 5. Current risk management for sea turtle disease hazards with notes on difficulties and defects.** A) Infectious diseases. B) Non-infectious diseases. C) One Health.

**A. Infectious diseases**

| Hazard | Current management | Difficulties and defects |
|---|---|---|
| Parasites: *Spirochiidae*, *Caryospora cheloniae*, *Ozobranchus branchiatus* Arthropod *spp.* | . Sporadic and opportunistic in rehabilitation | Data deficient, limited number of experts in this area, the diagnostic tests are not performed in many regional management units |
| Bacteria: *Vibrio spp.*, *Pseudomonas spp.*, *Escherichia coli*, *MRSA*, *Klebsiella* | Sporadic and opportunistic in rehabilitation. More recent research on antibiotic resistant bacteria | Data deficient, limited number of experts in this area, the diagnostic tests are not performed in many regional management units |
| Fungi: *Fusarium solani*, *Aspergillus spp.*, *Cladosporium spp.*, *Penicillium spp* | Sporadic and opportunistic in rehabilitation. Quarantine and hygiene in captivity | Data deficient, limited number of experts in this area, the diagnostic tests are not performed in many regional management units |
| Viruses: Chelonid alphaherpesvirus 5 (ChHV5) *Chelonia mydas* papillomavirus (CmPV-1) and *Caretta caretta* papillomavirus (CcPV-1): | Surgery in some regions, continuous research on epidemiology and aetiology | Data deficient, limited number of experts in this area, the diagnostic tests are not performed in many regional management units |

**B. Non-infectious diseases**

| Hazard | Current management | Difficulties and defects |
|---|---|---|
| Anthropogenic: Habitat degradation, Pollution/plastic | By-catch, accidents and entanglement: Marine park and governmental policies in some regions to use turtle exclusion devices (TED), and avoid stainless steel fishing hooks, avoid trawling. Defining protected areas to avoid accidents. Debris ingestion: Public involving workshops and programs to reduce plastic usage and littering near the ocean, and cleaning the beaches, rehabilitation | Region based, incompatible ethical and legal approaches across borders. |
| Environmental: Climate change | Debilitated Turtle Syndrome and cold stunning: Rehabilitation, training, educations | The capacity of rehabilitation is not enough in some regions with mass stranding; more research on treatment of specific conditions is required. |
| Medical | Malnutrition: Rehabilitation. Neurological diseases: managing toxin emissions in some areas | Neurological diseases: data deficiency. Lack of health baseline data |

**C. One Health**

| One Health consideration | Current management | Difficulties and defects |
|---|---|---|
| Zoonosis | Expanding the knowledge and awareness of meat and egg consumers | Sporadic reports |
| Bio-toxins | Expanding the knowledge and awareness of meat and egg consumers | Data deficient, mass death of humans, but no test to rule out contamination. Often in remote areas |
| Socioeconomic and cultural aspects of interacting with sea turtles | Expanding ecotourism and turtle watching activities. Implementing alternative jobs to avoid overfishing and poaching. Defining and modifying "*sustainable*" hunting for cultural purposes. Spiritual and cultural wellbeing of communities with close relationships to environment. Involving the communities in conservation programs. | Needs greater social science involvement |

*The global covid19 pandemic has had an adverse effect on ecotourism. Conservation organisations that depend on the funds from ecotourism are at risk of closure or financial crisis. Overfishing and poaching have been reported during lock-down. Conservationists are urged to rethink how to achieve their goals post-pandemic [141].

## 4. Discussion

Wildlife DRA as a decision-making tool is gaining recognition and DRA procedures and manuals have recently been published [6, 18]. However, there is no standardised and unified method to perform a DRA [41]. Workbooks, paired-ranking, expert workshops and scenario

**Table 6. Risk management options and scoring the effectiveness and feasibility in the Townsville management workshop.** A) Risk management options for Macroplastic pollution (effectiveness and feasibility reported from "1" the lowest to "10" the highest). B) Risk management options for Enterobacteriaceae and multi-resistant bacteria (effectiveness and feasibility reported from "1" the lowest to "10" the highest).

| 1) Reducing the risk of microplastic pollution | | | |
|---|---|---|---|
| **Management options** | **Effectiveness** | **Feasibility** | **Decision** |
| **eliminating the impacts of the macroplastic that has already been released** | | | |
| Initiatives to alter disposal methods | 7 | 5 | Beyond the scope of this group |
| Initiatives to clean beaches | 7 | 8 | Beyond the scope of this group |
| Installing storm drain filters | 9 | 7 | Beyond the scope of this group |
| Research on engineering structures to remove macroplastics from the ocean | 7 | 3 (due to cost) | Beyond the scope of this group |
| Government policies | 8 | 3 (political decision) | Beyond the scope of this group |
| **Reducing further input of macroplastic in the environment** | | | |
| Research on providing affordable biodegradable items | 7 | 3 | Beyond the scope of this group |
| Education and awareness to reduce littering and purchasing of plastics | 8 | 9 | This forms part of existing university subject curriculum, but needs to be addressed in primary and secondary schools as well. Not known to this group. Great Barrier Reef Marine Park Authority (GBRMPA) ReefHQ is an education facility and can educate on this topic as well. |
| Governmental policies | 8 | 3 | Beyond the scope of this group |
| **2) Reducing the risk of *Enterobacteriaceae* and multi-resistant bacteria** | | | |
| **Management options** | **Effectiveness** | **Feasibility** | **Decision** |
| Education and awareness including personal and protective equipment when working with sea turtles | 9 | 8 | This option is doable and already in practice. The Department of Environment and Science and GBRMPA have staff that would be involved in egg and turtle relocation and would require Personal protective equipment (PPE) as part of their risk assessment |
| Education and awareness to reduce the prescription and consumption of antibiotics | 6 | 8 | Beyond the scope of this group |
| Sewage treatment and extracting the antibiotics from sewage water | 7 | 3 | Beyond the scope of this group |

trees have been successfully used in previous analyses [6, 18, 37] and were therefore adapted in this study. The comprehensive explanation of each method is provided in the Jakob-Hoff *et al.*

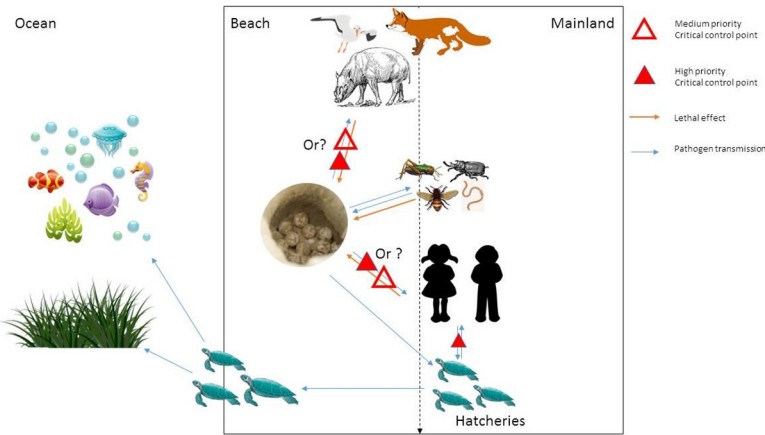

**Fig 5. The clutch translocation scenario, pathogen transmission pathways, lethal effects of predators and critical control points (the vectors, characters and icons in this figure were downloaded from the public domain https://pixabay.com/ and were modified according to its licence).**

[18]. The current study was an endeavour to update the information about health hazards of sea turtles in a structured way. Although, it is more practical to use a DRA for a specific scenario or case such as clutch translocation or hatchery establishment, this study provides up-to-date baseline information on a global scale and can serve as a guide to carry out such practices on a local scale.

Here, the hazard identification was more exhaustive than a standard review for DRA as it contains the collective information of disease causing hazards (S4-S8 Appendices in S1 File). The health hazards were assessed via a literature-based review and with input from experts in the field (Section 3). One of the considerable uncertainties revealed in this process was the data deficiency in the link between the presence of pathogens and infectious diseases of sea turtles. Additionally, viruses were identified as the least studied pathogens, although FP is suggested to have a viral aetiology. A higher rate of disease in immunocompromised individuals was repeatedly reported and a possible link between immunosuppression and environmental contaminants as a result of anthropogenic influence was suggested. One Health aspects, including the social element of interacting with sea turtles and society-based conservation, appeared to need more attention and research.

In this study, the risk management section was achieved through a global review of the current policies, possible management options and the difficulties of taking actions and was reviewed by members of IUCN Species Survival Commission (SSC) Sea turtle Specialist Group who are influential in making the policies and executing them.

This DRA is mainly a guide to support future risk assessments/management based on specific risk mitigating questions for which the management section should be done with the input of regional policy makers. Such discussions were initiated with appropriate local Australian government representatives to clarify appropriate steps in risk management for specific scenarios.

Conducting a DRA is an iterative process and risk analysis should continuously be reviewed and modified to represent the most recent information for policy and management decisions [41]. Disease surveillance and data collection to determine the contributing factors in population health is a practical approach to create evidence-based risk management actions for wildlife; and sea turtles are no exception. While future DRAs can benefit from this comprehensive review, the baseline information will undoubtedly expand as more pathogens are discovered, disease manifestations are reported and diagnostic tools are introduced.

The anthropogenic threats affecting sea turtles are increasing and so are the conservation initiatives to help these charismatic animals. Disease and health of sea turtles are not easily measured and management agencies are going to look for structured approaches to inform their decisions. The work presented here can form a platform for disease risk management of sea turtles, thereby aiding in their conservation.

## Supporting information

**S1 File.**
(DOCX)

## Acknowledgments

We would like to acknowledge the impacts that Dr. Andrew Barnes, Dr. Thierry Martin Work, and Dr. Mark Flint had on improving the manuscript.

## Author Contributions

**Conceptualization:** Narges Mashkour, Ellen Ariel.

**Data curation:** Narges Mashkour, Teresa Hipolito, Grant Walker, David Robinson, Warren Baverstock, Maryam Eghbali, Maryam Mohammadi, Daniela Freggi, Jane Giliam, Mike Hale, Nicholas Nicolle, Kevin Spiby, Daphne Wrobel, Mariluz Parga, Asghar Mobaraki, Rupika Rajakaruna, Kevin P. Hyland, Mark Read.

**Formal analysis:** Narges Mashkour, Shamim Ahasan, Richard Jakob-Hoff, Claire Saladin, Jose Luis Crespo-Picazo, Erina Young, David Blyde, Duan March.

**Funding acquisition:** Narges Mashkour.

**Methodology:** Karina Jones, Sara Kophamel, Mark Hamann, David Blyde, Duan March, Ellen Ariel.

**Project administration:** Narges Mashkour.

**Resources:** Narges Mashkour, Richard Jakob-Hoff.

**Supervision:** Narges Mashkour, Ellen Ariel.

**Validation:** Karina Jones, Sara Kophamel, Shamim Ahasan, Richard Jakob-Hoff, Maxine Whittaker, Mark Hamann, Ian Bell, Jennifer Elliman, Leigh Owens, Brett Gardner, Rachel Bowater, David Blyde, Duan March, Daphne Wrobel, Mariluz Parga, Ellen Ariel.

**Writing – original draft:** Narges Mashkour.

**Writing – review & editing:** Karina Jones, Sara Kophamel, Teresa Hipolito, Shamim Ahasan, Grant Walker, Richard Jakob-Hoff, Maxine Whittaker, Mark Hamann, Ian Bell, Jennifer Elliman, Leigh Owens, Claire Saladin, Jose Luis Crespo-Picazo, Brett Gardner, Aswini Leela Loganathan, Rachel Bowater, Erina Young, David Robinson, Maryam Eghbali, Maryam Mohammadi, Asghar Mobaraki, Ellen Ariel.

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
