## [Decision Letter · Decision Letter 0]

18 Jun 2020

PONE-D-20-06464

Disease Risk Analysis in sea turtles: A baseline study to inform conservation efforts

PLOS ONE

Dear Dr. Mashkour,

Thank you for submitting your manuscript to PLOS ONE. After careful consideration, we feel that it has merit but does not fully meet PLOS ONE’s publication criteria as it currently stands. Therefore, we invite you to submit a revised version of the manuscript that addresses the points raised during the review process.

While both reviewers agree that your study is has value, they also raise important issues. In particular, reviewer 1 provides a detailed list of items that must be addressed in order to make the manuscript acceptable for publication.

We look forward to receiving your revised manuscript.

Kind regards,

Ulrike Gertrud Munderloh, Ph.D.

Academic Editor

PLOS ONE

Journal Requirements:

Reviewers' comments:

Reviewer's Responses to Questions

**Comments to the Author**

1. Is the manuscript technically sound, and do the data support the conclusions?

Reviewer #1: Yes

Reviewer #2: Partly

2. Has the statistical analysis been performed appropriately and rigorously? 

Reviewer #1: N/A

Reviewer #2: N/A

3. Have the authors made all data underlying the findings in their manuscript fully available?

Reviewer #1: Yes

Reviewer #2: Yes

4. Is the manuscript presented in an intelligible fashion and written in standard English?

Reviewer #1: Yes

Reviewer #2: Yes

5. Review Comments to the Author

Reviewer #1: Please see attached pdf for reviewer comments.

Note: attached as pdf to preserve formatting.

......................................................................................................................................................................................

Reviewer #2: The manuscript reports an interesting Disease risk analysis (DRA) for sea turtles that is the result of an extensive study conducted by many authors through a systematic review and two workshops. The methodology appears solid and reliable and provides a guide for risk assessments to use in specific scenarios for conservation decisions about sea turtles. Despite the relevant information, the manuscript is however not currently suitable for publication, and it should be reviewed.

I understand the difficulties to summarize a so extensive study in a manuscript that follows the organization that it is required by a journal. However, in my opinion, the “materials and method” section is too long and some sentences sound as results or discussions. There is neither a “discussion” section nor a “conclusion” section, that is replaced with a summary. May be a discussion of the results and of the process that generate the DRA should be provided. The revision of the manuscript would make easier its reading.

6. PLOS authors have the option to publish the peer review history of their article (what does this mean?). If published, this will include your full peer review and any attached files.

Reviewer #1: Yes: David J. Duffy

Reviewer #2: No

---

## [Author Response · Author response to Decision Letter 0]

29 Jul 2020

Friday, July 17, 2020

Dear Editors,

The manuscript PONE-D-20-06464 entitled "Disease Risk Analysis in sea turtles: A baseline study to inform conservation efforts" has been revised in response to the comments from reviewer one and two.

We would like to thank the editorial board and the reviewers, especially reviewer one, for their constructive comments. We did our best to address their points and to revise the manuscript accordingly.

A table is provided and for each comment, our responses, the changes we made in the text and related sections/pages are clearly defined. The document is named “Response to Reviewers”.

The manuscript is revised and new references are added in the text and in reference list, hence “figure 3. The PRISMA chart” is also modified. Two manuscript files are attached one with track changes and one without.

Two very minor changes are made to the appendices which are addressed in “Response to Reviewers”.

We hope that the responses are in line with PLOS ONE’s publication criteria and would answer the reviewers’ questions.

Thanks for considering our manuscript for publication.

Sincerely,

Narges Mashkour

On behalf of the co-author team

Narges Mashkour (PhD.)

Teaching Assistant

Laboratory & Technical Support Scientist (Biomedicine)

College of Public Health, Medicine and Veterinary Sciences

James Cook University 

Townsville 4811, QLD, Australia

Ph: +61 420 336 741

---

## [Decision Letter · Decision Letter 1]

17 Sep 2020

Disease Risk Analysis in sea turtles: A baseline study to inform conservation efforts

PONE-D-20-06464R1

Dear Dr. Mashkour,

We’re pleased to inform you that your manuscript has been judged scientifically suitable for publication and will be formally accepted for publication once it meets all outstanding technical requirements.

Kind regards,

Ulrike Gertrud Munderloh, Ph.D.

Academic Editor

PLOS ONE

Additional Editor Comments (optional):

Reviewers' comments:

Reviewer's Responses to Questions

**Comments to the Author**

1. If the authors have adequately addressed your comments raised in a previous round of review and you feel that this manuscript is now acceptable for publication, you may indicate that here to bypass the “Comments to the Author” section, enter your conflict of interest statement in the “Confidential to Editor” section, and submit your "Accept" recommendation.

Reviewer #1: All comments have been addressed

Reviewer #2: All comments have been addressed

2. Is the manuscript technically sound, and do the data support the conclusions?

Reviewer #1: Yes

Reviewer #2: Yes

3. Has the statistical analysis been performed appropriately and rigorously? 

Reviewer #1: N/A

Reviewer #2: Yes

4. Have the authors made all data underlying the findings in their manuscript fully available?

Reviewer #1: Yes

Reviewer #2: Yes

5. Is the manuscript presented in an intelligible fashion and written in standard English?

Reviewer #1: Yes

Reviewer #2: Yes

6. Review Comments to the Author

Reviewer #1: The authors have implemented all of the suggested changes to the manuscript, or provided detailed rationale for changes they disagreed with. Therefore, I am happy to recommend the manuscript be accepted for publication in PLOS ONE.

Reviewer #2: The authors adequately addressed and discussed all comments. The data are all are fully available. The manuscript is correctly written, and the data support the conclusions. The manuscript should be accepted.

7. PLOS authors have the option to publish the peer review history of their article (what does this mean?). If published, this will include your full peer review and any attached files.

Reviewer #1: **Yes: **David Duffy

Reviewer #2: No

---

## [Editor Report · Acceptance letter]

21 Sep 2020

PONE-D-20-06464R1 

Disease Risk Analysis in sea turtles: A baseline study to inform conservation efforts 

Dear Dr. Mashkour:

I'm pleased to inform you that your manuscript has been deemed suitable for publication in PLOS ONE. Congratulations! Your manuscript is now with our production department. 

Kind regards, 

on behalf of

Dr. Ulrike Gertrud Munderloh 

Academic Editor

PLOS ONE